# FAM83H Expression Is Associated with Tumor-Infiltrating PD1-Positive Lymphocytes and Predicts the Survival of Breast Carcinoma Patients

**DOI:** 10.3390/diagnostics13182959

**Published:** 2023-09-15

**Authors:** Ji Eun Choi, Ae Ri Ahn, Junyue Zhang, Kyoung Min Kim, Ho Sung Park, Ho Lee, Myoung Ja Chung, Woo Sung Moon, Kyu Yun Jang

**Affiliations:** 1Department of Pathology, Chungnam National University Sejong Hospital, Sejong 30099, Republic of Korea; 2Department of Pathology, Jeonbuk National University Medical School, Jeonju 54896, Republic of Korea; xoxoyool@naver.com (A.R.A.); yuezai123@naver.com (J.Z.); kmkim@jbnu.ac.kr (K.M.K.); hspark@jbnu.ac.kr (H.S.P.); mjchung@jbnu.ac.kr (M.J.C.); mws@jbnu.ac.kr (W.S.M.); 3Research Institute of Clinical Medicine, Jeonbuk National University, Jeonju 54896, Republic of Korea; 4Research Institute, Jeonbuk National University Hospital, Jeonju 54896, Republic of Korea; 5Department of Forensic Medicine, Jeonbuk National University Medical School, Jeonju 54896, Republic of Korea; foremed@jbnu.ac.kr

**Keywords:** breast, carcinoma, immunohistochemistry, FAM83H, PD1, prognosis

## Abstract

Background: FAM83H has been implicated in cancer progression, and PD1 is an important target for anti-cancer immune checkpoint therapy. Recent studies suggest an association between FAM83H expression and immune infiltration. However, studies on the roles of FAM83H and its relationship with PD1 in breast carcinomas have been limited. Methods: Immunohistochemical expression of FAM83H and PD1 and their prognostic significance were evaluated in 198 breast carcinomas. Results: The expression of FAM83H in cancer cells was significantly associated with the presence of PD1-positive lymphoid cells within breast carcinoma tissue. Individual and co-expression patterns of nuclear FAM83H and PD1 were significantly associated with shorter survival of breast carcinomas in univariate analysis. In multivariate analysis, the expression of nuclear FAM83H (overall survival, *p* < 0.001; relapse-free survival, *p* = 0.003), PD1 (overall survival, *p* < 0.001; relapse-free survival, *p* = 0.003), and co-expression patterns of nuclear FAM83H and PD1 (overall survival, *p* < 0.001; relapse-free survival, *p* < 0.001) were the independent indicators of overall survival and relapse-free survival of breast carcinoma patients. Conclusions: This study suggests a close association between FAM83H expression and the infiltration of PD1-positive lymphoid cells in breast carcinomas and their expression as the prognostic indicators for breast carcinoma patients, and further studies are needed to clarify this relationship.

## 1. Introduction

The family with sequence similarity 83 member H (FAM83H) has a major role in teeth development [1,2] and cytoskeletal assembly [3,4]. In addition, cancer genomic analysis for the FAM83H family indicates that the FAM83H gene is increased in various types of human cancers, such as breast, lung, liver, and colorectal cancers [5]. Moreover, the role of FAM83H in maintaining cytoskeletal integrity necessitates investigating its roles in cancer progression, especially regarding the epithelial to mesenchymal transition (EMT) [3,4,6]. Furthermore, higher expression of FAM83H was significantly associated with shorter survival of cancer patients in the liver [7], bone [8], stomach [6], kidney [9], pancreas [10], and colorectum [11]. In human cancers, FAM83H is involved in cancer progression in conjunction with MYC [7], the Wnt/β-catenin pathway [6,8], and the PI3K/AKT pathway [10,12]. Therefore, FAM83H has been suggested as an important molecular target of human cancers [6,7,8,9,10].

Immune evasion of cancer cells is one of the important hallmarks of cancer progression which may continuously increase until causing the death of patients [13]. Among the immune evasion mechanisms of cancer cells, the interaction between programmed cell death 1 (PD1) and programmed death-ligand 1 (PD-L1) is critical and important in cancer treatment because immune checkpoint inhibitors targeting PD1/PD-L1 show therapeutic efficacy [13,14]. As a cancer marker, PD1 expression was significantly associated with the survival of various human cancers [15,16,17,18]. However, despite the therapeutic efficacy of anti-PD1 therapy in some populations of cancer patients, there was also a population of cancer patients who did not repose to anti-PD1 therapy [13,19]. Therefore, an explanation of the mechanism of how cancer is sustained by escaping immune surveillance during anti-cancer therapies is needed [14,19,20].

Breast cancer is the most commonly diagnosed female cancer [21]. Despite advances in cancer treatment stratagem, breast cancer is the leading cause of female cancer death worldwide [21]. Recently, there have been extensive trials regarding immune checkpoint blockage targeting the PD1/PD-L1 pathway in cancer therapy [14,20,22]. Nevertheless, due to the resistance observed in patients receiving immune checkpoint inhibitors, there is a growing need to identify alternative biomarkers that are strongly linked to therapy resistance [19,20]. These biomarkers can be utilized to predict the response to checkpoint inhibitors and enhance the effectiveness of such treatments [13,14,20,22]. Recently, FAM83H has been suggested as a possible therapeutic target of human cancers [6,7,8], and it has been suggested that FAM83H might be involved in the regulation of immune infiltration in cancer tissue [10,23]. The Human Protein Atlas (https://www.proteinatlas.org) database also shows that higher expression of FAM83H mRNA is associated with shorter survival of breast cancer patients (Accession date; 13 July 2023) [24]. However, studies focused on the roles of FAM83H in conjunction with immune infiltration in breast carcinoma (BCA) are limited. Therefore, this study investigated the expression and prognostic significance of FAM83H and intra-tumoral infiltration of PD1-positive lymphoid cells in human BCAs.

## 2. Materials and Methods

### 2.1. BCA Patients

This study included BCA patients who underwent surgery between January 1997 and December 2003. A total of 198 cases were selected based on the availability of medical records, histologic slides, and paraffin-embedded tissue blocks. Clinicopathological information was obtained by reviewing medical records and slides. Of the 198 patients, 180 received postoperative chemotherapy, and 157 received postoperative hormone therapy. None of the patients received neoadjuvant chemotherapy. The histopathologic factors and tumor stage were reviewed according to the WHO classification of breast tumors [21] and the American Joint Committee Cancer Staging System [25]. The clinicopathological factors evaluated in the study included age, tumor stage, T category, lymph node metastasis, distant metastatic relapse, histologic type, histologic grade, and three characteristics of histologic grading (tubule and glandular formation, nuclear pleomorphism, and mitotic count). The study also evaluated the immunohistochemical expression status of HER2, estrogen receptor, and progesterone receptor, which were obtained from the pathology reports. This study was approved by the institutional review board of Jeonbuk National University Hospital (IRB number CUH 2023-01-008) and was conducted in accordance with the Declaration of Helsinki. Informed consent was waived due to its anonymous and retrospective nature.

### 2.2. Immunohistochemical Staining and Scoring

The expression of FAM83H and PD1 in BCA tissues was evaluated with immunohistochemical staining using tissue microarray (TMA) tissue sections. The TMAs were constructed from the original paraffin-embedded tissue blocks from the area with the highest histologic grade, mainly composed of tumor cells. Two 3.0 mm-sized cores were constructed in each case. The tissue sections from the TMA were deparaffinized and underwent antigen retrieval by boiling for 20 min in pH 6.0 antigen retrieval buffer (DAKO, Glostrup, Denmark) using a microwave oven. The tissue sections were incubated with primary antibodies for FAM83H (Cat# A304-323A, 1:100, Bethyl Laboratories, Montgomery, TX, USA) and PD1 (Cat# ab52587, clone NAT105, 1:50, Abcam, Cambridge, UK) and then visualized using the DAKO Envision system (DAKO, Carpinteria, CA, USA). Immunohistochemical expression of FAM83H was separately evaluated for its nuclear and cytoplasmic expression based on the staining intensity (a scale of 0 to 3: 0, no staining; 1, weak staining; 2, intermediate staining; 3, strong staining) and staining area (a scale of 0 to 5: 0, 0%; 1, 1%; 2, 2–10%; 3, 11–33%; 4, 34–66%; 5, 67–100%) [11,26,27]. The immunohistochemical staining score was obtained by adding a staining intensity scale and staining area scale for each TMA core. Thereafter, the final immunohistochemical score was obtained by summing the scores of two TMA cores, and the final score ranged from zero to sixteen. Immunohistochemical staining slides for PD1 were evaluated by counting the number of tumor-infiltrating PD1-positive cells at the highest-numbered five high-power fields in each TMA core [16]. The counting was performed under a Nikon ECLIPSE 80i light microscope (Nikon, Tokyo, Japan) with an ×10 eyepiece with a 25 mm field of view (Nikon, Tokyo, Japan) and ×40 objective lens (Plan Flour ×40/0.75; Nikon, Tokyo, Japan). The field diameter was 0.625 mm, and the area of one high power field (HPF) was 0.3068 mm^2^. The total area analyzed per case was 3.068 mm^2^. The slides were evaluated by two pathologists (KYJ and HSP), who reached a consensus score through simultaneous observation. The entire process was performed blind to the clinicopathological information.

### 2.3. Statistical Analysis

The immunohistochemical positivity of FAM83H and PD1 expression was evaluated using receiver operating characteristic curve analysis [11,15,28]. The cut-off points for the immunohistochemical staining scores for FAM83H and PD1 were determined based on the point with the highest area under the curve to predict the death of patients with BCA by December 2014 [28,29]. The prognostic significance of FAM83H and PD1 expression in BCAs was assessed for overall survival (OS) and relapse-free survival (RFS). For OS analysis, the survival duration was calculated from the date of operation to the date of death or last contact. The event considered in the OS analysis was the death of patients from BCA. Patients who were still alive or had died from other causes at the end of the follow-up were censored. For RFS analysis, an event was defined as either relapse or death without relapse of patients from BCA. The survival duration for RFS analysis was calculated from the date of operation to the date of event or last contact. Patients who were either alive without recurrence or had died from other causes at the end of the follow-up were censored. Survival analysis was conducted via univariate and multivariate Cox proportional hazards regression analysis as well as Kaplan–Meier survival analysis. Factors that were significant in the univariate analysis were included in the multivariate analysis. Pearson’s chi-square test and Student’s *t*-test were used to determine the association between clinicopathological factors. Statistical analysis was performed using SPSS software (version 22.0, IBM, Armonk, NY, USA) with a *p*-value of less than 0.05 considered statistically significant.

## 3. Results

### 3.1. The Association between the Clinicopathologic Variables and the Expression of FAM83H and PD1 in BCAs

In BCA tissue, FAM83H was expressed in both the nuclei and cytoplasm of tumor cells, and PD1 was expressed in tumor-infiltrating lymphoid cells (Figure 1A). Positivity for the immunohistochemical expression of nuclear FAM83H (FAM83H-Nu), cytoplasmic FAM83H (FAM83H-Cy), and PD1 was determined using receiver operating characteristic curve analysis to predict the death of patients from BCA. The cut-off points for FAM83H-Nu, FAM83H-Cy, and PD1 were 11, 9, and 22, respectively (Figure 1B). Cases with an immunohistochemical staining score equal to or greater than 11 for FAM83H-Nu and 9 for FAM83H-Cy were considered positive for FAM83H-Nu and FAM83H-Cy, respectively (Figure 1B). Cases containing equal to or greater than 22 PD1-positive cells in ten high-power fields from two TMA cores were classified as PD1-positive (Figure 1B). With these cut-off values, FAM83H-Nu expression was significantly associated with age (*p* = 0.042), tumor stage (*p* = 0.008), lymph node metastasis (*p* = 0.007), distant metastatic relapse (*p* < 0.001), histologic grade (*p* = 0.007), tubule and gland formation (*p* = 0.029), nuclear pleomorphism (*p* = 0.002), and the expression of HER2 (*p* < 0.001), ER (*p* = 0.009), PR (*p* = 0.034), FAM83H-Cy (*p* < 0.001), and PD1 (*p* < 0.001) (Appendix A). The number of PD1-positive cells was significantly higher in the FAM83H-Nu positive group (mean ± standard error, 44.8 ± 6.5) as compared to the FAM83H-Nu negative group (mean ± standard error, 24.6 ± 4.8) (*p* = 0.012) (Appendix A). FAM83H-Cy positivity was significantly associated with distant metastatic relapse (*p* = 0.029), histologic grade (*p* < 0.001), tubule and glandular formation (*p* = 0.007), nuclear pleomorphism (*p* < 0.001), and the expression of HER2 (*p* = 0.002), ER (*p* = 0.020), PR (*p* = 0.013), and PD1 (*p* = 0.009) (Appendix A). PD1 positivity was significantly associated with distant metastatic relapse (*p* < 0.001) and histologic grade (*p* = 0.012) (Appendix A).

### 3.2. The Expression of FAM83H and PD1 Are Associated with Shorter Survival of BCA Patients

The significant factors associated with OS or RFS in univariate analysis were age (OS, *p* < 0.001; RFS, *p* = 0.032), tumor stage (OS, *p* = 0.021; RFS, *p* = 0.097), T category of stage (OS, *p* = 0.038; RFS, *p* = 0.197), lymph node metastasis (OS, *p* = 0.049; RFS, *p* = 0.159), histologic grade (OS, *p* = 0.008; RFS, *p* = 0.151), mitotic count (OS, *p* = 0.014; RFS, *p* = 0.270), and the expression of HER2 (OS, *p* = 0.009; RFS, *p* = 0.025), PR (OS, *p* = 0.093; RFS, *p* = 0.005), FAM83H-Nu (OS, *p* < 0.001; RFS, *p* < 0.001), FAM83H-Cy (OS, *p* < 0.001; RFS, *p* < 0.001), and PD1 (OS, *p* < 0.001; RFS, *p* < 0.001) (Table 1). BCA patients positive for FAM83H-Nu had an 11.633-fold greater risk of death (95% confidence interval (95% CI), 4.966–27.253) and a 6.475-fold higher risk of death or cancer relapse (95% CI, 3.670–11.424) compared to BCA patients that were FAM83H-Nu negative (Table 1). FAM83H-Cy positivity predicted a 4.303-fold greater risk of death (95% CI, 2.160–8.572) and a 3.284-fold higher risk of death or cancer relapse (95% CI, 1.933–5.580) (Table 1). BCA patients who were positive for PD1 had a 5.194-fold higher risk of death (95% CI, 2.779–9.706) and a 3.188-fold higher risk of death or cancer relapse (95% CI: 1.993–5.099) compared to BCA patients who were negative for PD1 (Table 1). Kaplan–Meier survival curves for OS and RFS according to the expression of FAM83H-Nu, FAM83H-Cy, and PD1 are presented in Figure 2.

Multivariate analysis was conducted using factors that showed a significant association with OS or RFS in the univariate analysis. The factors included in the multivariate analysis were age, tumor stage, T category of the stage, lymph node metastasis, histologic grade, mitotic count, and the expression of HER2, PR, FAM83H-Nu, FAM83h-Cy, and PD1. In multivariate analysis, the expression of FAM83H-Nu and PD1 were independent indicators of OS and RFS of BCA patients (Table 2). FAM83H-Nu-positive BCA patients had an 8.098-fold greater risk of death (95% CI, 3.409–19.237) and a 5.268-fold higher risk of death or cancer relapse (95% CI, 2.939–9.441) than patients who were negative for FAM83H-Nu (Table 2). BCA patients who were positive for PD1 had a 3.220-fold higher risk of death (95% CI, 1.707–6.075) and a 2.095-fold higher risk of death or cancer relapse (95% CI: 1.292–3.395) compared to patients who were negative for PD1 (Table 2).

### 3.3. Co-Expression Patterns of Nuclear FAM83H and PD1 Predict Survival of BCA Patients

We further evaluated the prognostic significance of FAM83H and PD1 expression in subgroups of BCA patients according to positivity for FAM83H-Nu and PD1. In the 115 PD1-negative patients, the expression of both FAM83H-Nu (OS, *p* < 0.001; RFS, *p* < 0.001) and FAM83H-Cy (OS, *p* < 0.001; RFS, *p* < 0.001) were significantly associated with OS and RFS (Figure 3A). FAM83H-Nu positivity was significantly associated with OS (*p* < 0.001) and RFS (*p* < 0.001) in the 83 patients who were positive for PD1 (Figure 3B). PD1 expression was also significantly associated with OS and RFS in both FAM83H-Nu-negative (OS, *p* = 0.002; RFS, *p* = 0.003) and FAM83H-Nu-positive (OS, *p* = 0.002; RFS, *p* = 0.045) subgroups (Figure 3C,D).

In multivariate analysis, the expression of FAM83H-Nu and PD1 were the independent indicators of shorter OS and RFS of BCA patients. In addition, there was a significant association between FAM83H-Nu positivity and PD1 positivity (Appendix A). Therefore, we have evaluated the prognostic significance in four subgroups of BCA patients according to the co-expression pattern of FAM83H-Nu and PD1: FAM83H-Nu^−^/PD1^−^, FAM83H-Nu^−^/PD1^+^, FAM83H-Nu^+^/PD1^−^, and FAM83H-Nu^+^/PD1^+^. In these four subgroups, the FAM83H-Nu^−^/PD1^−^ subgroup showed the longest survival (5-year OS rate, 100%; 10-year OS rate, 99%; 5-year RFS rate, 96%; 10-year RFS rate, 92%), and the FAM83H-Nu^+^/PD1^+^ subgroup had the shortest survival (5-year OS rate, 63%; 10-year OS rate, 48%; 5-year RFS rate, 44%; 10-year RFS rate, 28%) (Table 3) (Figure 4A). The FAM83H-Nu^−^/PD1^+^ and FAM83H-Nu^+^/PD1^−^ subgroups showed intermediate survival duration (Table 3) (Figure 4A), and there was no significant prognostic difference between the FAM83H-Nu^−^/PD1^+^ and FAM83H-Nu^+^/PD1^−^ subgroups (Figure 4A). Based on these results, we subclassified BCA patients into three prognostic subgroups according to the co-expression pattern of FAM83H-Nu and PD1: [FAM83H-Nu^−^/PD1^−^], [FAM83H-Nu^−^/PD1^+^ or FAM83H-Nu^+^/PD1^−^], and [FAM83H-Nu^+^/PD1^+^]. This subgrouping was a significant indicator of OS and RFS of BCA patients (Table 4) (Figure 4B). These co-expression patterns of FAM83H-Nu and PD1 predicted OS and RFS in univariate (OS, *p* < 0.001; RFS, *p* < 0.001) and multivariate analysis (OS, *p* < 0.001; RFS, *p* < 0.001) (Table 4).

### 3.4. Prognostic Significance of the Expression of FAM83H and PD1 in Subgroups of BCAs Based on Adjuvant Therapies and in Triple-Negative BCAs

We further evaluated the prognostic significance of FAM83H and PD1 expression in subgroups of BCA patients based on adjuvant therapies and in triple-negative (ER^−^/PR^−^/HER2^−^) BCAs. In 180 BCA patients who received adjuvant chemotherapy, the expression of FAM83H-Nu (OS, *p* < 0.001; RFS, *p* < 0.001), FAM83H-Cy (OS, *p* < 0.001; RFS, *p* < 0.001), and PD1 (OS, *p* < 0.001; RFS, *p* < 0.001) and the co-expression patterns of FAM83H-Nu and PD1 (OS, overall *p* < 0.001; RFS, overall *p* < 0.001) were significantly associated with OS and RFS (Figure 5A). In 157 BCA patients who received adjuvant hormone therapy, the expression of FAM83H-Nu (OS, *p* < 0.001; RFS, *p* < 0.001), FAM83H-Cy (OS, *p* < 0.001; RFS, *p* < 0.001), and PD1 (OS, *p* < 0.001; RFS, *p* < 0.001) and co-expression patterns of FAM83H-Nu and PD1 (OS, overall *p* < 0.001; RFS, overall *p* < 0.001) were significantly associated with OS and RFS (Figure 5B). In triple-negative BCAs, the expression of FAM83H-Nu (*p* = 0.010) and FAM83H-Cy (*p* = 0.024) and the co-expression patterns of FAM83H-Nu and PD1 (overall *p* = 0.032) were significantly associated with OS (Figure 5C). The expression of FAM83H-Nu (*p* = 0.032) was significantly associated with the RFS of triple-negative BCAs (Figure 5C).

## 4. Discussion

In this study, FAM83H was expressed in both the nuclei and cytoplasm of tumor cells in BCA tissue, and the expression of both FAM83H-Nu and FAM83H-Cy were significantly associated with the survival of BCA patients in univariate analysis. Especially, the expression of FAM83H-Nu showed significant associations with various clinicopathologic variables related to cancer progression, such as a higher tumor stage and histologic grade, the presence of lymph node metastasis, and a distant metastatic relapse. Furthermore, the expression of FAM83H-Nu was an independent indicator of shorter OS and RFS of BCA patients. With regards to the expression of FAM83H in human cancer cells, the expression of FAM83H was higher in cancer tissue compared with normal counterpart tissue [5,12,30]. In addition, in line with this study, the nuclear expression of FAM83H predicted a shorter survival of cancer patients compared with its cytoplasmic expression [6,9,11,31]. Nuclear expression of FAM83H was an independent indicator of survival of cancer patients in gastric [6], gallbladder [31], kidney [9], and colorectal cancers [11]. These findings suggest that the subcellular localization of FAM83H and its expression patterns might have implications for its functional roles in cancer progression. However, in osteosarcoma tissue, the prognostic impact of FAM83H-Cy was more potent than FAM83H-Nu, and FAM83H-Cy was an independent prognostic indicator in multivariate analysis [8]. Therefore, the prognostic significance of FAM83H expression according to its nuclear and cytoplasmic localization might vary according to cancer type. However, when considering the effects of FAM83H overexpression on the stimulation of proliferation and invasiveness of cancer cells [6,7,8], it is suggested that higher levels of FAM83H expression, and consequently its localization in the nuclei, might play a role in cancer progression. The possible explanation for the roles of FAM83H in cancer progression is closely related to the activation of the β-catenin pathway [6,8]. In gastric carcinoma cells, FAM83H is involved in cancer progression by inducing EMT through the stabilization of β-catenin, preventing it from undergoing proteasomal degradation [6]. In osteosarcoma cells, a higher expression of FAM83H was associated with the nuclear localization of β-catenin and subsequent activation of the β-catenin-related signaling pathway [8]. In addition, during the stabilization of the β-catenin pathway, FAM83H interacts with adhesion molecules such as SCRIB and Nectin1 to stimulate cellular proliferation and promote EMT-mediated invasiveness [6,32]. FAM83H-mediated cancer progression related to EMT-related cancer metastasis has also been shown in an in vivo osteosarcoma model [8]. Consistently, FAM83H-Nu positivity was significantly associated with clinical factors related to cancer invasiveness, such as lymph node metastasis and latent distant metastasis of BCAs in this study. In addition, FAM83H expression was transcriptionally controlled by MYC in liver cancer cells [7], and FAM83H overexpression was significantly associated with KRAS mutation and EMT gene signature in pancreatic cancer [10]. In pancreatic and uterine cervical cancer cells, FAM83H was found to be involved in cancer progression through the activation of the PI3K/Akt pathway [10,12]. Therefore, in addition to its usability as a prognostic indicator of BCAs, this study suggests that FAM83H might play a role in the progression of BCAs in association with various oncogenic signaling pathways.

In addition to the prognostic role of FAM83H expression in BCA patients, intra-tumoral infiltration of PD1-positive cells was also significantly associated with the survival of BCA patients. Multivariate analysis indicated PD1 positivity as an independent indicator of OS and RFS of BCA patients. Consistently, it has been reported that a higher infiltration of intra-tumoral PD1-positive cells was significantly associated with shorter OS and disease-free survival of BCAs [17,18]. A higher number of PD1-positive T-follicular helper T cells in sentinel lymph nodes were also associated with shorter survival of triple-negative BCAs [33]. Higher expression of PD1 mRNA in peripheral blood was associated with shorter OS, but, in contrast, higher PD1 mRNA in tumor tissue was associated with favorable OS in BCAs [34]. In addition, PD1 protein positivity in tumor-infiltrating lymphocytes was associated with favorable OS and disease-free survival of triple-negative BCAs [35,36]. Therefore, because there is controversy on the prognostic significance of PD1 expression in BCAs, further studies are needed to clarify the clinicopathological significance of intra-tumoral infiltration of PD1-positive lymphoid cells.

Furthermore, another interesting finding of this study is the significant association between FAM83H expression and intra-tumoral infiltration of PD1-positive lymphoid cells. The possible correlation between FAM83H expression and anti-tumor immunity was suggested in the study of pancreatic cancer patients [10]. In pancreatic cancer, overexpression of FAM83H was associated with decreased infiltration of intra-tumoral CD8+ cytotoxic T cells and shorter survival of patients [10]. Similarly, in gastric cancer, higher expression of FAM83H was significantly associated with decreased infiltration of B cells, CD4+ helper T cells, CD8+ T cells, macrophages, and myeloid dendritic cells [23]. In line with these findings, this study showed that the FAM83H-Nu-positive group had a significantly higher number of PD1-positive cells compared to the FAM83H-Nu-negative group. Moreover, both FAM83H-Nu positivity and PD1 positivity were independent indicators of shorter OS and RFS of BCA patients in multivariate analysis. Notably, co-positivity for FAM83H-Nu and PD1 predicted the shortest OS and RFS in BCA patients. These findings suggest that FAM83H expression, especially in the nuclei of tumor cells, might be associated with the immune environment of BCA. Consistently, when we searched the TIMER2.0 database (Accession date; 13 July 2023, http://timer.cistrome.org), higher gene expression of FAM83H was associated with an increased infiltration of regulatory T cells and a decreased infiltration of myeloid dendritic cells in BCAs [37], suggesting immune evasion in BCAs with high FAM83H expression. However, studies on the role of FAM83H in anti-tumor immunity have been limited. Therefore, further study is needed to clarify the roles of FAM83H in the immune evasion of cancer cells.

In the treatment of BCAs, various adjuvant therapeutic strategies are used, depending on the biomolecular characteristics of the BCAs. When we evaluated the prognostic significance of FAM83H and PD1 expression in the subpopulations of BCAs according to adjuvant therapies, individual and co-expression patterns of FAM83H-Nu and PD1 were significantly associated with the OS and RFS of patients who received adjuvant chemotherapy and hormone therapy. Therefore, the prognostic significance of PD1 expression in this study suggests potential for anti-PD1 therapy as a therapeutic strategy for BCA patients. However, there are controversial results on the efficacy of anti-PD1/anti-PD-L1 therapy in BCA patients. Especially in triple-negative BCAs, the therapeutic efficacy of anti-PD1 or anti-PD-L1 monotherapy remains suboptimal and even lower than in other solid tumors, ranging from 5% to 23% [22]. Consequently, there is a growing focus on investigating combination approaches, such as chemotherapy and other immune checkpoint inhibitors, and, currently, these strategies outnumber single immune checkpoint inhibitor trials [38]. In our results, despite the limited number of patients, FAM83H-Nu positivity and the co-expression patterns of FAM83H and PD1 were significantly associated with the shorter OS of 32 triple-negative BCAs. Moreover, in addition to the positive association between FAM83H-Nu positivity and PD1 positivity, FAM83H-Nu was significantly associated with shorter survival in both PD1-negative and PD1-positive subgroups of BCAs. In addition, PD1 positivity is also significantly associated with shorter OS and RFS in both the FAM83H-Nu-negative and FAM83H-Nu-positive subgroups of BCAs. These results might be related to independent roles of FAM83H and PD1 in the progression of human cancers, especially the roles of FAM83H in cancer progression through cellular proliferation and activation of EMT through the canonical WNT/β-catenin pathway [6,7,8]. Therefore, this study suggests that a combination therapy targeting both FAM83H and PD1 might be used in the treatment of BCA patients.

## 5. Conclusions

In conclusion, this study demonstrated a close association between the expression of FAM83H in tumor cells and the presence of tumor-infiltrating PD1-positive lymphoid cells in BCAs. Furthermore, the individual and co-expression patterns of FAM83H-Nu and PD1 were significantly associated with OS and RFS of BCA patients. Especially, FAM83H-Nu positivity predicted survival of PD1-negative and PD1-positive subgroups of BCAs and vice versa. Therefore, this study suggests that the expression of FAM83H and PD1 might be used as both prognostic indicators and novel therapeutic targets for BCA patients. However, further studies are needed to clarify the efficacy of the FAM83H expression and intra-tumoral infiltration of PD1-positive lymphoid cells as prognostic indicators and novel therapeutic targets in BCA patients.

## Figures and Tables

**Figure 1 diagnostics-13-02959-f001:**
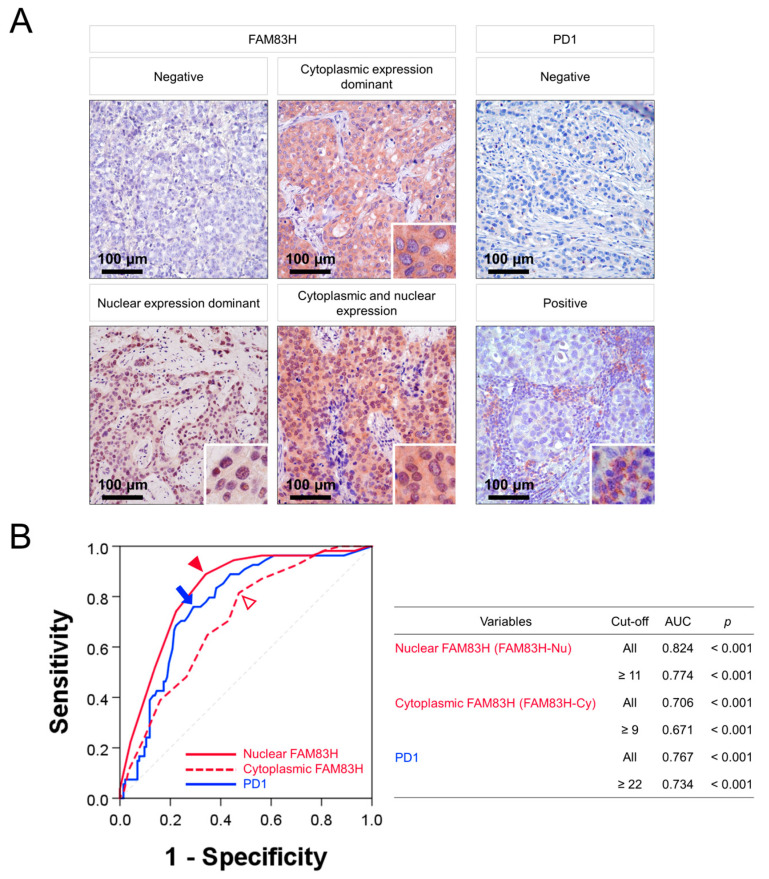
Immunohistochemical expression of FAM83H and PD1 in breast carcinomas and statistical analysis. (**A**) FAM83H is expressed in both the cytoplasm and nuclei of breast carcinoma cells. PD1 is expressed in tumor-infiltrating lymphoid cells. Original magnification: ×400. (**B**) Receiver operating characteristic curve analysis to determine cut-off points for the expression of nuclear FAM83H (FAM83H-Nu, red arrowhead), cytoplasmic FAM83H (FAM83H-Cy, empty red arrowhead), and PD1 (blue arrow). The cut-off points represent the point of highest area under the curve (AUC) to predict the death of patients with breast carcinoma.

**Figure 2 diagnostics-13-02959-f002:**
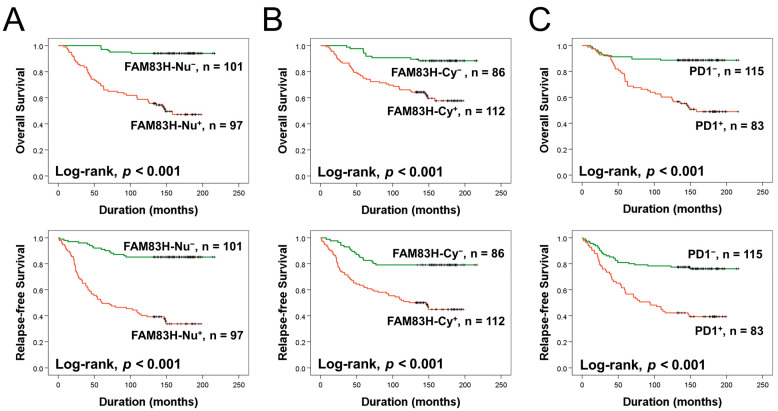
Survival analysis of overall survival and relapse-free survival in breast carcinoma patients based on the expression of nuclear and cytoplasmic FAM83H and PD1. Kaplan–Meier survival curves according to the expression of nuclear FAM83H (FAM83H-Nu) (**A**), cytoplasmic FAM83H (FAM83H-Cy) (**B**), and PD1 (**C**).

**Figure 3 diagnostics-13-02959-f003:**
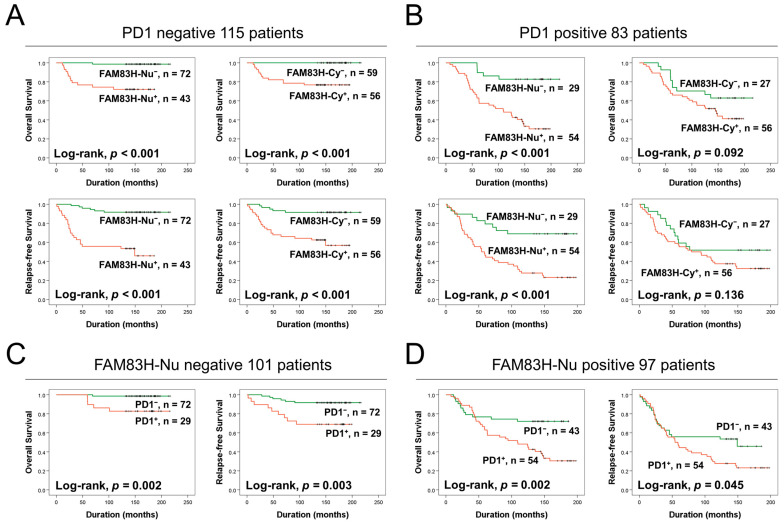
Survival analysis of overall survival and relapse-free survival in subpopulations of breast carcinomas according to the positivity of PD1 and nuclear expression of FAM83H. Kaplan–Meier survival curves for nuclear FAM83H (FAM83H-Nu) and cytoplasmic FAM83H (FAM83H-Cy) in the subpopulations of PD1-negative (**A**) and PD1-positive (**B**) breast carcinomas. Kaplan–Meier survival curves for PD1 positivity in the subpopulations of FAM83H-Nu-negative (**C**) and FAM83H-Nu-positive (**D**) breast carcinomas.

**Figure 4 diagnostics-13-02959-f004:**
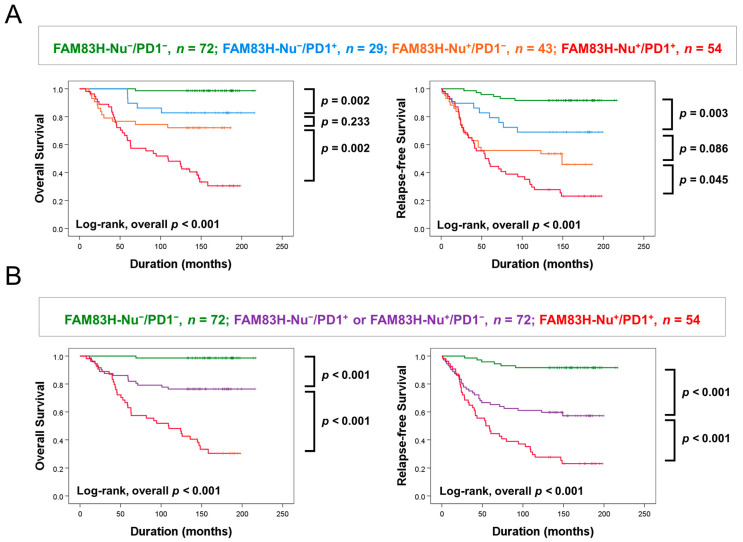
Survival analysis according to the co-expression patterns of nuclear FAM83H and PD1. (**A**) Kaplan–Meier survival curves for four subgroups of breast carcinomas according to the expression of nuclear FAM83H (FAM83H-Nu) and PD1: FAM83H-Nu^−^/PD1^−^, FAM83H-Nu^−^/PD1^+^, FAM83H-Nu^+^/PD1^−^, and FAM83H-Nu^+^/PD1^+^ subgroups. (**B**) Kaplan–Meier survival curves in three subgroups of breast carcinomas: [FAM83H-Nu^−^/PD1^−^], [FAM83H-Nu^−^/PD1^+^ or FAM83H-Nu^+^/PD1^−^], and [FAM83H-Nu^+^/PD1^+^] subgroups.

**Figure 5 diagnostics-13-02959-f005:**
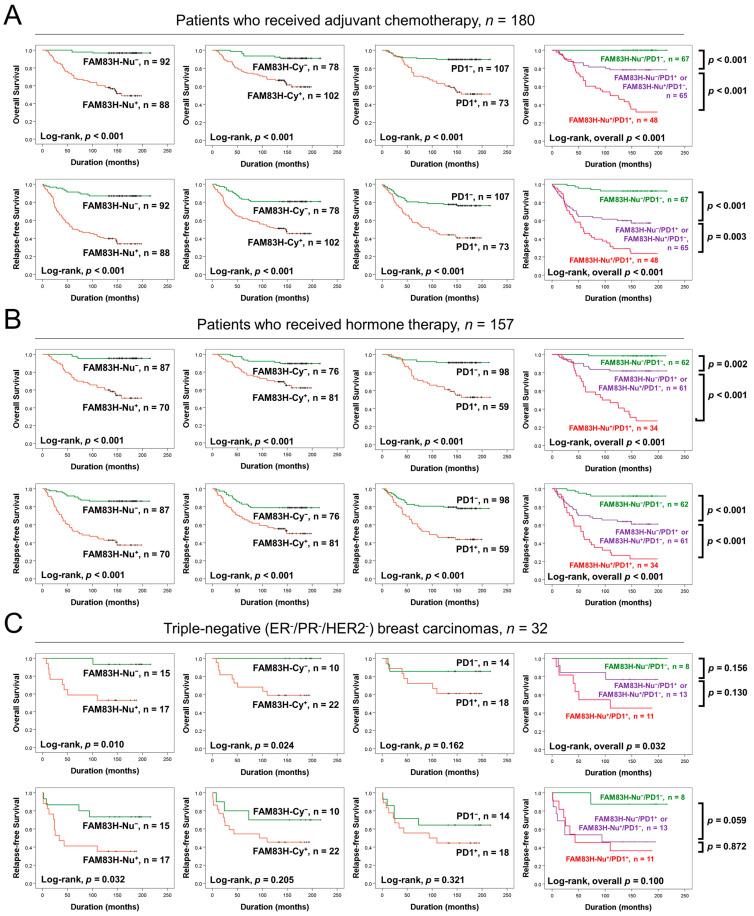
Survival analysis in subgroups of breast carcinoma patients based on adjuvant therapies and in triple-negative breast carcinomas. Kaplan–Meier survival curves according to the expression of nuclear FAM83H (FAM83H-Nu), cytoplasmic FAM83H (FAM83H-Cy), PD1, and co-expression patterns of FAM83H-Nu and PD1 in the subpopulations of breast carcinoma patients who received adjuvant chemotherapy (**A**) and hormone therapy (**B**) as well as in triple-negative (ER^−^/PR^−^/HER2^−^) breast carcinomas (**C**).

**Table 1 diagnostics-13-02959-t001:** Univariate Cox regression analysis of overall survival and relapse-free survival in BCAs.

Characteristics		No.	OS		RFS	
			HR (95% CI)	*p*	HR (95% CI)	*p*
Age, y	≥50 (vs. <50)	62/198	2.473 (1.449–4.219)	<0.001	1.649 (1.045–2.602)	0.032
Tumor stage	I	40/198	1	0.021	1	0.097
	II	122/198	2.408 (0.942–6.156)	0.067	1.591 (0.826–3.064)	0.165
	III and IV	36/198	4.065 (1.476–11.195)	0.007	2.278 (1.075–4.825)	0.032
T category of stage	1	62/198	1	0.038	1	0.197
	2	120/198	2.542 (1.233–5.239)	0.011	1.580 (0.933–2.676)	0.089
	3 and 4	16/198	2.562 (0.859–7.647)	0.092	1.769 (0.743–4.210)	0.197
Lymph node metastasis	presence (vs. absence)	91/198	1.717 (1.001–2.946)	0.049	1.379 (0.882–2.157)	0.159
Histologic type	NST (vs. lobular)	190/198	0.866 (0.211–3.559)	0.842	1.744 (0.704–4.321)	0.230
Histologic grade	1	55/198	1	0.008	1	0.151
	2	96/198	0.904 (0.449–1.817)	0.776	1.062 (0.606–1.863)	0.833
	3	47/198	2.247 (1.124–4.493)	0.022	1.674 (0.916–3.056)	0.094
Tubule and gland formation	1	31/198	1	0.662	1	0.371
	2	70/198	1.192 (0.498–2.854)	0.694	1.390 (0.651–2.968)	0.394
	3	97/198	1.419 (0.621–3.241)	0.406	1.651 (0.803–3.393)	0.172
Nuclear pleomorphism	1	11/168	1	0.227	1	0.640
	2	69/198	0.522 (0.172–1.587)	0.252	0.998 (0.346–2.877)	0.997
	3	118/198	0.876 (0.312–2.461)	0.801	1.249 (0.450–3.461)	0.670
Mitoses/10 HPFs	0–9	127/198	1	0.014	1	0.270
	10–19	33/198	1.208 (0.570–2.561)	0.621	0.977 (0.518–1.845)	0.944
	>19	38/198	2.429 (1.329–4.440)	0.004	1.526 (0.894–2.605)	0.121
HER2	positive (vs. negative)	65/198	2.038 (1.193–3.481)	0.009	1.679 (1.067–2.644)	0.025
ER	positive (vs. negative)	132/198	0.640 (0.372–1.102)	0.107	0.659 (0.417–1.042)	0.074
PR	positive (vs. negative)	109/198	0.632 (0.370–1.079)	0.093	0.523 (0.333–0.822)	0.005
FAM83H-Cy	positive (vs. negative)	112/198	4.303 (2.160–8.572)	<0.001	3.284 (1.933–5.580)	<0.001
FAM83H-Nu	positive (vs. negative)	97/198	11.633 (4.966–27.253)	<0.001	6.475 (3.670–11.424)	<0.001
PD1	positive (vs. negative)	83/198	5.194 (2.779–9.706)	<0.001	3.188 (1.993–5.099)	<0.001

Abbreviations: OS, overall survival; RFS, relapse-free survival; HR, hazard ratio; 95% CI, 95% confidence interval; HPFs, high-power fields; FAM83H-Nu, nuclear FAM83H; FAM83H-Cy, cytoplasmic FAM83H; NST, no special type; ER, estrogen receptor; PR, progesterone receptor.

**Table 2 diagnostics-13-02959-t002:** Multivariate Cox regression analysis of overall survival and relapse-free survival in BCAs.

Characteristics		OS		RFS	
		HR (95% CI)	*p*	HR (95% CI)	*p*
Age, y	≥50 (vs. <50)	1.840 (1.076–3.147)	0.026		
FAM83H-Nu	positive (vs. negative)	8.098 (3.409–19.237)	<0.001	5.268 (2.939–9.441)	<0.001
PD1	positive (vs. negative)	3.220 (1.707–6.075)	<0.001	2.095 (1.292–3.395)	0.003

Multivariate survival analysis was performed with the factors significantly associated with OS or RFS in univariate analysis: age, tumor stage, T category of tumor stage, lymph node metastasis, histologic grade, mitotic count, and the expression of HER2, ER, PR, nuclear FAM83H, cytoplasmic FAM83H, and PD1. Abbreviations: OS, overall survival; RFS, relapse-free survival; HR, hazard ratio; 95% CI, 95% confidence interval; FAM83H-Nu, nuclear FAM83H.

**Table 3 diagnostics-13-02959-t003:** Five- and ten-year overall survival and relapse-free survival according to co-expression patterns of nuclear FAM83H and PD1 in BCAs.

Co-Expression Pattern of FAM83H-Nu and PD1	No.	5y-OS (%)	10y-OS (%)	5y-RFS (%)	10y-RFS (%)
Co-Expression Model 1					
FAM83H-Nu^−^/PD1^−^	72	100	99	96	92
FAM83H-Nu^−^/PD1^+^	29	90	83	79	69
FAM83H-Nu^+^/PD1^−^	43	77	72	56	56
FAM83H-Nu^+^/PD1^+^	54	63	48	44	28
Co-expression Model 2					
FAM83H-Nu^−^/PD1^−^	72	100	99	96	92
FAM83H-Nu^−^/PD1^+^ or FAM83H-Nu^+^/PD1^−^	72	82	76	65	61
FAM83H-Nu^+^/PD1^+^	54	63	48	44	28

Abbreviations: 5y-OS, five-year overall survival rate; 10y-OS, ten-year overall survival rate; 5y-RFS, five-year relapse-free survival rate; 10y-RFS, ten-year relapse-free survival rate; FAM83H-Nu/PD1, co-expression pattern of nuclear FAM83H and PD1.

**Table 4 diagnostics-13-02959-t004:** Univariate and multivariate Cox regression analysis of overall survival and relapse-free survival according to co-expression patterns of nuclear FAM83H and PD1 in BCAs.

Characteristics		No.	OS		RFS	
			HR (95% CI)	*p*	HR (95% CI)	*p*
Univariate analysis						
FAM83H-Nu/PD1	FAM83H-Nu^−^/PD1^−^	72/198	1	<0.001	1	<0.001
	FAM83H-Nu^−^/PD1^+^ or FAM83H-Nu^+^/PD1^−^	72/198	20.686 (2.752–155.493)	0.003	6.685 (2.780–16.073)	<0.001
	FAM83H-Nu^+^/PD1^+^	54/198	72.789 (9.962–531.859)	<0.001	14.932 (6.316–35.305)	<0.001
Multivariate analysis						
Age, y	≥50 (vs. <50)		1.847 (1.080–3.156)	0.025		
FAM83H-Nu/PD1	FAM83H-Nu^−^/PD1^−^		1	<0.001	1	<0.001
	FAM83H-Nu^−^/PD1^+^ or FAM83H-Nu^+^/PD1^−^		19.365 (2.573–145.721)	0.004	6.685 (2.780–16.073)	<0.001
	FAM83H-Nu^+^/PD1^+^		65.801 (8.984–481.957)	<0.001	14.932 (6.316–35.305)	<0.001

Multivariate survival analysis was performed with the factors significantly associated with OS or RFS in univariate analysis: age, tumor stage, T category of tumor stage, lymph node metastasis, histologic grade, mitotic count, HER2 expression, ER expression, PR expression, and co-expression patterns of nuclear FAM83H and PD1. Abbreviations: OS, overall survival; RFS, relapse-free survival; HR, hazard ratio; 95% CI, 95% confidence interval, FAM83H-Nu/PD1, co-expression pattern of nuclear FAM83H and PD1.

## Data Availability

The datasets generated during and/or analyzed during the current study are available from the corresponding author on reasonable request.

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
