# Peer review of "FAM83H Expression Is Associated with Tumor-Infiltrating PD1-Positive Lymphocytes and Predicts the Survival of Breast Carcinoma Patients"

_diagnostics, 2023, doi:10.3390/diagnostics13182959_

Round 1
Reviewer 1 Report
The authors analysed by immunohistochemical the expression of FAM83H and PD1 in 189 breast cancer samples and present a series of prognostic and correlation analyses. It is a well written manuscript with clear aims, although it is necessary to revise some English oversights. The methods used in the study and the results are well described and discussed.
I suggest to put table 1 as supplementary material, and I would be more cautious about the conclusions, stating that “further studies are needed “ to use FAM83H and PD1 expression as prognostic indicators and novel therapeutic targets in breast cancer patients.
Author Response
Response to reviewer 1
We thank the reviewer for the insightful comments.
Comments to Author:
Reviewer #1:
The authors analysed by immunohistochemical the expression of FAM83H and PD1 in 189 breast cancer samples and present a series of prognostic and correlation analyses. It is a well written manuscript with clear aims, although it is necessary to revise some English oversights. The methods used in the study and the results are well described and discussed.
I suggest to put table 1 as supplementary material, and I would be more cautious about the conclusions, stating that “further studies are needed “ to use FAM83H and PD1 expression as prognostic indicators and novel therapeutic targets in breast cancer patients.
We thank the reviewer for this comment. In response to the reviewer’s comment, we have revised to present Table 1 as Supplementary Table S1. In addition, we have revised our conclusion to state that “further studies are needed”. Below are sentences revised in the conclusions.
Abstract section
This study suggests a close association between FAM83H expression and the infiltration of PD1-positive lymphoid cells in breast carcinomas and their expression as the prognostic indicators for breast carcinoma patients, and further studies are needed to clarify this relationship.
In conclusion section
Therefore, this study suggests that the expression of FAM83H and PD1 might be used as both prognostic indicators and novel therapeutic targets for BCA patients. However, further studies are needed to clarify the efficacy of the FAM83H expression and intra-tumoral infiltration of PD1-positive lymphoid cells as prognostic indicators and novel therapeutic targets in BCA patients.
Reviewer 2 Report
The authors analyzed immunohistochemical expression of FAM83H which is expressed in many human cancers and PD1 (programmed cell death) in 198 breast cancer samples.
The expression of both FAM83H-Nu and –Cy was significantly associated with survival of BCA patients in univariable analysis. The expression of FAM83H-Nu was independent indicator of shorter OS and RFS. It showed significant associations with higher tumor stage, histologic grade, presence of lymph node metastasis, and distant metastatic relapse. Significant association between FAM83H expression and intra-tumoral infiltration of PD1 – positive lymphoid cells. Survival and relapse-free survival analysis of the breast cancer patients divided into subgroups according to the positive or negative expression of both genes under study revealed that expression of FAM83H or PD1 was always connected with the worse survival or relapse-free survival. For the combination of both genes, patients with FAM83H-/PD1- had the best survival by contrast to the patients with FAM83H+/PD1+ who had the worst survival rate.
The manuscript is interesting. The No of the patients under study is acceptable.
Author Response
Response to reviewer 2
We thank the reviewer for the insightful comments.
Comments to Author:
Reviewer #2:
The authors analyzed immunohistochemical expression of FAM83H which is expressed in many human cancers and PD1 (programmed cell death) in 198 breast cancer samples.
The expression of both FAM83H-Nu and –Cy was significantly associated with survival of BCA patients in univariable analysis. The expression of FAM83H-Nu was independent indicator of shorter OS and RFS. It showed significant associations with higher tumor stage, histologic grade, presence of lymph node metastasis, and distant metastatic relapse. Significant association between FAM83H expression and intra-tumoral infiltration of PD1 – positive lymphoid cells. Survival and relapse-free survival analysis of the breast cancer patients divided into subgroups according to the positive or negative expression of both genes under study revealed that expression of FAM83H or PD1 was always connected with the worse survival or relapse-free survival. For the combination of both genes, patients with FAM83H-/PD1- had the best survival by contrast to the patients with FAM83H+/PD1+ who had the worst survival rate.
The manuscript is interesting. The No of the patients under study is acceptable.
We very thank the reviewer for this comment.